# Kynurenine Pathway of Tryptophan Metabolism Is Associated with Hospital Mortality in Patients with Acute Respiratory Distress Syndrome: A Prospective Cohort Study

**DOI:** 10.3390/antiox11101884

**Published:** 2022-09-23

**Authors:** Li-Chung Chiu, Hsiang-Yu Tang, Chun-Ming Fan, Chi-Jen Lo, Han-Chung Hu, Kuo-Chin Kao, Mei-Ling Cheng

**Affiliations:** 1Department of Thoracic Medicine, Chang Gung Memorial Hospital, Chang Gung University College of Medicine, Taoyuan 33305, Taiwan; 2Graduate Institute of Clinical Medical Sciences, College of Medicine, Chang Gung University, Taoyuan 33302, Taiwan; 3Metabolomics Core Laboratory, Healthy Aging Research Center, Chang Gung University, Taoyuan 33302, Taiwan; 4Clinical Metabolomics Core Laboratory, Chang Gung Memorial Hospital, Taoyuan 33305, Taiwan; 5Department of Respiratory Therapy, Chang Gung Memorial Hospital, Chang Gung University College of Medicine, Taoyuan 33305, Taiwan; 6Department of Respiratory Therapy, Chang Gung University College of Medicine, Taoyuan 33302, Taiwan; 7Department of Biomedical Sciences, College of Medicine, Chang Gung University, Taoyuan 33302, Taiwan

**Keywords:** acute respiratory distress syndrome, metabolic profiling, kynurenine, tryptophan, oxidative stress, mortality

## Abstract

Acute respiratory distress syndrome (ARDS) involves dysregulated immune-inflammatory responses, characterized by severe oxidative stress and high mortality. Metabolites modulating the inflammatory and immune responses may play a central role in the pathogenesis of ARDS. Most biogenic amines may induce the production of reactive oxygen species, oxidative stress, mitochondrial dysfunction, and programmed cell death. We conducted a prospective study on metabolic profiling specific to the amino acids and biogenic amines of 69 patients with ARDS. Overall, hospital mortality was 52.2%. Between day 1 and day 7 after ARDS onset, plasma kynurenine levels and the kynurenine/tryptophan ratio were significantly higher among non-survivors than in survivors (all *p* < 0.05). Urine metabolic profiling revealed a significantly higher prevalence of tryptophan degradation and higher concentrations of metabolites downstream of the kynurenine pathway among non-survivors than among survivors upon ARDS onset. Cox regression models revealed that plasma kynurenine levels and the plasma kynurenine/tryptophan ratio on day 1 were independently associated with hospital mortality. The activation of the kynurenine pathway was associated with mortality in patients with ARDS. Metabolic phenotypes and modulating metabolic perturbations of the kynurenine pathway could perhaps serve as prognostic markers or as a target for therapeutic interventions aimed at reducing oxidative stress and mortality in ARDS.

## 1. Introduction

Acute respiratory distress syndrome (ARDS) is a heterogeneous syndrome with alveolar injury secondary to clinical insults, involving inflammatory cascades and immune responses. ARDS can lead to life-threatening refractory hypoxemia and/or multiple organ failure, with mortality reaching 50% in severe cases [1,2]. Currently, there are no reliable biomarkers by which to predict disease progression or guide the application of therapies for ARDS. Lung-protective mechanical ventilation remains the cornerstone of ARDS management [3].

Metabolomics refers to the high-throughput characterization of metabolites that play crucial roles in cellular physiology, inflammation, and the activation of immune cells by modulating the genome, epigenome, transcriptome, or proteome. Metabolic derangement provides an objective means by which to identify biological endotypes, in terms of molecular mechanisms and pathogenesis [4,5,6]. An association between metabolism and inflammatory or immune responses (i.e., metabolite-cytokine correlations) has also been observed in patients with coronavirus disease 2019 (COVID-19) [7,8,9]. Metabolomics can be used to assess the risk of developing ARDS and facilitate patient stratification, predict the severity or progression, and predict responses to drugs or other medical interventions (i.e., pharmacometabolomics). However, the results of recent metabolomic studies in ARDS have not been consistent [4,10,11]. It is believed that metabolomics could eventually be used as a therapeutic target by which to enhance the precision of strategies for the treatment of ARDS [4,10,11,12,13,14].

Most of the biogenic amines form through the decarboxylation of amino acids and could induce the production of reactive oxygen species (ROS), oxidative stress via the amine oxidase-mediated catabolic pathway, mitochondrial dysfunction, and programmed cell death [15,16,17]. Kynurenine is a toxic metabolite of biogenic amines that forms through the breakdown of tryptophan, an essential amino acid obtained entirely via dietary intake. The tryptophan-kynurenine pathway is involved in the regulation of inflammation, immunity, neuronal function, and intestinal homeostasis. This pathway tends to be accelerated in cases of infectious and inflammatory insults or immune response activation. It has also been implicated in a variety of diseases and pathological conditions, such as autoimmune disorders, cancer, and neurodegenerative diseases [15,18,19,20,21,22,23].

The extensive ROS production contributing to oxidative stress can increase the permeability of the pulmonary endothelial bed and plays an important role in the pathogenesis and progression of ARDS [24]. The activation of the kynurenine pathway was linked with increased oxidative stress, while kynurenine can exert immunosuppressive and anti-inflammatory effects via the ROS pathway [18,19,20,21,22,23]. Note that researchers have yet to elucidate the impact of the kynurenine pathway on the progression of ARDS, clinical outcomes, or the mortality of ARDS patients. Our objective in this prospective study was to use metabolic profiling that is specific to amino acids and biogenic amines, not a broad omics study, to examine the association between serial changes in metabolic profiles and hospital mortality among patients with ARDS.

## 2. Materials and Methods

### 2.1. Study Design and Patient Cohort

This prospective study on the metabolic profiling of amino acids and biogenic amines involved 69 ARDS patients and was conducted from February 2017 to June 2018, in the medical ICUs at a tertiary care referral center, the Chang Gung Memorial Hospital in Taiwan. ARDS was defined in accordance with the Berlin criteria [25]. Written informed consent was obtained from all patients who participated in the study. The exclusion criteria included the following: age < 20 years, potentially confounding comorbidities, multiple organ failure refractory to therapy (i.e., a moribund condition), death within 3 days after ARDS onset, and failure to obtain informed consent. During the initial phase of ARDS, all the patients were deeply sedated and paralyzed. Mechanical ventilator settings were collected during the use of a neuromuscular blockade. Inclusion criteria for healthy control subjects were sex- and age-matched residents who were living at Chang Gung Health and Culture Village in Taiwan. Exclusion criteria for healthy controls were those persons who had evidence of significant comorbidities. The local Institutional Review Board for Human Research approved this study (IRB No. 201407524B0, 201801052A3, and 201801497B0).

### 2.2. Data Collection

Demographic data, the etiologies of ARDS, and major comorbidities were recorded. Arterial blood gas and mechanical ventilator settings were recorded at approximately 10 a.m. on day 1 after the onset of ARDS, as well as on day 3 and day 7. Clinical and laboratory variables were recorded, and sequential organ failure assessment (SOFA) scores were calculated at day 1, day 3, and day 7. We also collected the dates of hospital and ICU admission, mechanical ventilator initiation and liberation, ARDS diagnosis, ICU and hospital discharge, and time of death. Mechanical power was calculated using the following equation: mechanical power (Joules/minutes) (J/min) = 0.098 × tidal volume × respiratory rate × (peak inspiratory pressure − 1/2 × driving pressure) [26,27]. Hospital mortality refers to all causes of death during the hospital stay.

### 2.3. Sample Collection and Preparation

Plasma and urine samples were obtained from patients after fasting overnight (for at least 8 h) at day 1, day 3, and day 7 after ARDS onset. Plasma and urine samples were also obtained from the healthy controls under the same dietary constraints.

Whole blood (10 mL) was collected under sterile conditions using an EDTA-coated tube, which was then centrifuged within 2 h (3000 rpm; 4 °C; 10 min of activity), whereupon the plasma was transferred into a 15 mL polypropylene tube. To separate cell debris, the plasma was centrifuged again at 3000 rpm at 4 °C for 10 min. After the second centrifugation, the plasma was aliquoted in an Eppendorf tube and stored at −80 °C until it was assayed. Urine samples were centrifuged at 3000 rpm at 4 °C for 10 min with the supernatant collected in an Eppendorf tube and stored at −80 °C until it was assayed (Figure 1 and the Appendix A).

### 2.4. Amino Acids and Biogenic Amine Analysis of Plasma and Urine Samples

Amino acids and biogenic amines were measured via ultra-high performance liquid chromatography and liquid chromatography-mass spectrometry, respectively. The urine samples were diluted with distilled water to achieve a creatinine content of 100 μg/mL, as measured using an ultra-high performance liquid chromatography system. The relevant details pertaining to the methods used in sample processing are provided (see the Appendix A).

### 2.5. Statistical Analysis

Continuous variables are presented as mean and standard deviations for normally distributed variables or median and interquartile range for non-normally distributed variables. Student’s *t*-test was used for the comparison of normally distributed data, and a Mann–Whitney U-test was used for the nonparametric data. Categorical variables were presented as frequencies and percentages and were compared, using the chi-squared test for equal proportions or Fisher’s exact test. The receiver-operating characteristic curve and the Youden index were used to determine the cutoff, to dichotomize the continuous variables. Univariate analysis was used to identify the risk factors associated with hospital mortality, followed by the construction of Cox proportional hazard regression models with stepwise selection. The results are presented as hazard ratios (HR) with 95% confidence intervals (CI). The probability of survival was analyzed using the Kaplan–Meier method and compared between groups, using the log-rank test. Statistical analysis was performed using SPSS Statistics, version 26.0, and statistical significance was considered to be represented by a two-tailed *p-*value of less than 0.05.

## 3. Results

### 3.1. Study Patients

A total of 69 patients with ARDS and 30 healthy controls were enrolled for this prospective metabolic profiling cohort study. The mean age of healthy controls and ARDS patients was 66.3 and 65.2 years (*p* = 0.532), respectively. Twenty-four healthy persons (80%) and 53 patients (76.8%) were male in terms of the healthy controls and ARDS patients (*p* = 0.726). The main cause of ARDS was bacterial pneumonia (*n* = 50, 72.5%), followed by influenza pneumonia (*n* = 7, 10.1%). There were no significant comorbidities among healthy controls; 9 persons (30%) had hypertension and 3 persons (10%) had diabetes mellitus. An immunocompromised status (32 patients, 46.4%) was the most common comorbidity in ARDS patients, followed by hypertension (24 patients, 34.8%). Thirty-six patients with ARDS died, and the overall hospital mortality was 52.2% (Table 1). The most common cause of death in ARDS patients was sepsis, with multiple organ failure (*n* = 24, 66.7%) followed by respiratory failure (*n* = 7, 19.4%), an underlying immunocompromised status (*n* = 4, 11.1%), and profound cardiogenic shock (*n* = 1, 2.8%).

Heat maps and volcano plot analysis revealed distinct quantitative differences between survivors and non-survivors, in terms of the metabolic phenotyping of amino acids and biogenic amines (Figure 2). The quantitative metabolic profiling of amino acids and the biogenic amines of healthy controls, survivors, and non-survivors (from day 1 to day 7 after ARDS onset) was provided. The comparisons of values between healthy controls and ARDS patients, and between survivors and non-survivors, are shown in Appendix A. Note that plasma kynurenine was the only metabolite with significantly higher values in non-survivors than in survivors (from day 1 to day 7 after ARDS onset).

### 3.2. Comparisons of Survivors and Non-Survivors

As shown in Table 1, there were no significant differences between survivors and non-survivors in terms of age, sex, or ARDS etiology. Body weight and body mass index were significantly higher among survivors than among non-survivors. A higher proportion of non-survivors were immunocompromised.

The SOFA scores at day 1 were significantly higher among non-survivors than among survivors, and these values decreased between day 1 and day 7 among survivors but increased among non-survivors. At day 1, plasma kynurenine levels were significantly higher among patients than among healthy controls (25.1 ± 19.1 vs. 3.7 ± 0.6 µM, *p* < 0.001). Among the non-survivors, between day 1 and day 7, we observed a stepwise increase in the mean concentration of plasma kynurenine. Among the survivors, between day 1 and day 7, we observed a stepwise decrease in the mean kynurenine/tryptophan ratio. On all sampling days, plasma kynurenine levels and the plasma kynurenine/tryptophan ratio were both significantly higher among non-survivors than among survivors (all *p* < 0.05) (Table 1 and Figure 2). Non-survivors had significantly lower PaO_2_/FiO_2_ values and were treated with significantly higher FiO_2_ than survivors at day 3 and day 7 after the onset of ARDS (all *p* < 0.05). There were no significant differences between the two groups in terms of ventilator settings at day 1. Baseline (day 1) plasma kynurenine levels were significantly correlated with SOFA scores (Figure 3).

### 3.3. Comparing Patients with High and Low Plasma Kynurenine Levels

For the early identification of biomarkers to predict the outcomes of patients with ARDS, the maximum Youden index value was used to categorize patients according to plasma kynurenine levels at day 1, using a cutoff of 15.12 µM, dividing them into the high plasma kynurenine group (45 patients; 65%) and the low plasma kynurenine group (24 patients; 35%). As shown in Table 2, no significant differences were observed between the high plasma kynurenine group and the low plasma kynurenine group, in terms of age, sex, body weight, or body mass index. Between day 1 and day 7, there was a stepwise decrease in SOFA scores in the low kynurenine group. The SOFA scores were significantly lower in the low kynurenine group than in the high plasma kynurenine group (all *p* < 0.05). Between day 1 and day 7, the plasma kynurenine/tryptophan ratios were significantly higher in the high plasma kynurenine group than in the low plasma kynurenine group. There were no significant differences between the two groups in terms of ventilator settings at day 1. The incidence of hospital mortality was significantly higher in the high plasma kynurenine group than in the low plasma kynurenine group (66.7% vs. 25%, *p* = 0.003).

### 3.4. Urine Metabolite Profiling: Tryptophan Degradation

The metabolic profiles of the urine samples at day 1 revealed a significantly higher incidence of tryptophan degradation among non-survivors than among survivors (fold change 0.72; *p* = 0.031). All downstream kynurenine pathway metabolites were higher among non-survivors (day 1 and day 3) than among survivors (day 1 of ARDS onset), and many of the differences reached significance (Table 3). Figure 4 presents a schematic illustration showing the kynurenine pathway of tryptophan catabolism, with the altered metabolites highlighted with red arrows.

### 3.5. Factors Associated with Hospital Mortality

After adjusting for significant confounding variables, Cox proportional hazard regression models revealed a number of factors that were independently associated with an elevated risk of hospital mortality: immunocompromised status, the SOFA score at day 1, plasma lactate level at day 1, plasma kynurenine level at day 1, and plasma kynurenine/tryptophan ratio at day 1. Hazard of death estimates that were obtained using the plasma kynurenine/tryptophan ratio at day 1 were higher than estimates obtained using plasma kynurenine at day 1 (adjusted HR = 1.761 and 1.017, respectively; both *p* < 0.05). Plasma kynurenine levels of >15.12 µM at day 1 were independently associated with higher hospital mortality (adjusted HR = 4.317 [95% CI 1.621–11.495]; *p* = 0.003) (Table 4). The overall hospital survival rate was significantly higher among patients with lower plasma kynurenine values on day 1 (≤15.12 µM) than among those with higher plasma kynurenine values on day 1 (>15.12 µM) (75% vs. 33.3%; *p* = 0.003; log-rank test) (Figure 5).

## 4. Discussion

Our main insight in this prospective metabolic profiling study of amino acids and biogenic amines was that at the time of ARDS onset, plasma kynurenine levels and the plasma kynurenine/tryptophan ratio were both independently associated with the likelihood of hospital mortality. Our findings indicate that the activation of the kynurenine pathway may play a role in pathogenesis and may, therefore, serve as a biomarker by which to predict clinical outcomes in cases of ARDS.

Kynurenine can exert immunosuppressive effects via the aryl hydrocarbon receptor, which suppresses the proliferation of effector T cells and natural killer cells and promotes the activation of regulatory T cells. Increased kynurenine levels have been shown to alter the cellular metabolism, exert anti-inflammatory effects, and cause cell death via the ROS pathway [15,19,20,21,22,23,28]. Under the effects of an inflammatory stimulus or immune system activation, proinflammatory cytokines and chemokines (particularly tumor necrosis factor-*α* and interferon-*γ* via activated T-cells and the T helper type 1 cell-mediated immune response) can enhance the activity of indoleamine 2,3-dioxygenase (IDO), which is a negative regulator of inflammation and immunization. IDO, which is highly expressed in antigen-presenting cells (e.g., dendritic cells), catalyzes the first and main rate-limiting step in the kynurenine pathway. In this way, it contributes to the breakdown of tryptophan and the accumulation of kynurenine, with corresponding effects on immunosuppression and immune tolerance [18,19,20,21,22,28,29].

A recent study was conducted by Metwaly et al. to identify the metabolic fingerprints in ARDS and its subgroups (direct and indirect ARDS) and subphenotypes (hypoinflammatory and hyperinflammatory ARDS) [30]. It included ARDS patients (*n* = 108; median age 59 years; 28-day mortality, 21.3%) and a control group (ICU-ventilated patients; *n* = 27; median age 57 years; 28-day mortality, 12.5%). Most ARDS cases were secondary to pneumonia and sepsis, similar to our study cohort. In the longitudinal tracking of metabolite changes in the recovery group of ARDS (*n* = 43) at days 1, 7, 14, and 28 after ICU admission, the data revealed that the levels of several metabolites appeared to move toward control levels within 7–14 days following ICU admission, which coincided with the time of clinical improvement. Kynurenine reached a peak value at day 7 and slowly declined later on in ARDS patients but remained higher than in the control group at day 1 (fold change 9.9; *p* = 0.003), day 7, day 14, and day 28 after ICU admission. However, the above study did not evaluate the association between serial changes of metabolites and mortality.

In the current study, we enrolled healthy subjects as the control group (*n* = 30; mean age 66.3 years) and ARDS patients (*n* = 69; mean age 65.2 years; hospital mortality 52.2%). We evaluated the association between serial changes in metabolic profiles and hospital mortality among patients with ARDS; we did not compare the metabolic profiles of ARDS subgroups and subphenotypes, as in the previous study [30]. We also found that plasma kynurenine levels were significantly higher among ARDS patients on all sampling days (day 1, 3, and 7) than among the control group (ARDS patients at day 1 vs. healthy control: 25.1 ± 19.1 vs. 3.7 ± 0.6 µM, *p* < 0.001). A stepwise increase in the mean concentration of plasma kynurenine was found among non-survivors from day 1 to day 7; plasma kynurenine was the only metabolite for which the mean value remained significantly higher among non-survivors than among survivors throughout the study period. In our Cox regression model, plasma kynurenine values at day 1 were independently associated with hospital mortality (HR = 1.017, *p* = 0.017). Urine metabolic profiling revealed higher tryptophan degradation and higher downstream kynurenine pathway metabolites among non-survivors at day 1, indicating the activation of the kynurenine pathway. However, the concentration of each urine metabolite was normalized to each urine sample’s corresponding creatinine level, to compensate for urine volume variations [31]. Twenty ARDS patients (29%) had chronic kidney disease and some ARDS patients experienced acute kidney injury, with elevated creatinine levels, oliguria, or even anuria in our study. This may make urine metabolite analysis less practical than serum metabolite analysis and limit the number of patients available for statistical analysis. Therefore, urine metabolites were not used to assess the relationship with hospital mortality in this study.

The kynurenine-to-tryptophan ratio has been widely used to estimate the enzyme activity of IDO, where an elevated kynurenine/tryptophan ratio often indicates that the activation of IDO correlates with elevated neopterin levels (an indicator of cellular immune activation and oxidative stress) [8,9,18,22,32,33,34]. Kynurenine is a neurotoxic metabolite; the activation of the kynurenine pathway (assessed in terms of plasma kynurenine levels and the kynurenine/tryptophan ratio) was independently associated with acute brain dysfunction (delirium and coma) in mechanically ventilated patients [35]. Sepsis is the leading cause of ARDS; one previous study reported that the kynurenine/tryptophan ratio was up to ninefold and was significantly higher in patients with septic shock than in the two control groups (non-septic, low blood pressure controls, and normotensive healthy subjects). The kynurenine/tryptophan ratio was strongly correlated with inotrope requirements [32]. The kynurenine pathway has also been implicated in tumor-associated immunosuppression, wherein IDO may promote the evasion of tumor cells from the immune system’s surveillance. The overexpression of IDO is associated with poor prognosis in a variety of cancers, and clinical trials on IDO inhibitors for cancer immunotherapy are currently underway [19,21,29,36].

Researchers have not previously investigated the potential role of the kynurenine/tryptophan ratio or IDO activity in cases of ARDS. Higher kynurenine levels were found in ARDS patients, but the inverse is not observed for tryptophan in this study. The kynurenine/tryptophan ratio in this study is not as elevated as previously reported in earlier studies. This may be related to the half-life of kynurenine and tryptophan, variations in tryptophan-containing parenteral infusions, and the tryptophan metabolism pathways (protein biosynthesis and the serotonin and kynurenine pathways) among patients with ARDS [19]. Our findings revealed that between day 1 and day 7, the mean kynurenine/tryptophan ratio (an index for IDO activity) was significantly lower among survivors than among non-survivors and that the ratio decreased over time. Cox regression models revealed that estimates of the hazard of death obtained using the plasma kynurenine/tryptophan ratio at day 1 were higher than those estimates obtained using plasma kynurenine at day 1, despite the fact that both factors were independently associated with hospital mortality (adjusted HR = 1.761 and 1.017, respectively; both *p* < 0.05). This is an indication that inflammatory signals during ARDS enhanced the IDO activity, which subsequently activated the kynurenine pathway, thereby participating in pathogenesis and disease progression, with a corresponding effect on clinical outcomes. It is possible that the kynurenine/tryptophan ratio or IDO activity could be used as a prognostic tool for patient stratification or for the development of drugs aimed at improving the clinical outcomes of ARDS [29].

Recent studies described strong associations between metabolites, including the kynurenine pathway of tryptophan metabolism and proinflammatory cytokines/chemokines (e.g., interleukin (IL)-1 and IL-6) in COVID-19 patients [7,8,37]. Targeting the tryptophan metabolism was shown to modulate the release of proinflammatory cytokines via peripheral blood mononuclear cells isolated from rhesus macaques that were infected ex vivo with severe acute respiratory syndrome coronavirus 2 (SARS-CoV-2) [7]. These results indicate that by intervening in metabolic dysregulation, it may be possible to suppress the release of cytokines in COVID-19 patients. The kynurenine/tryptophan ratio was also significantly increased among patients infected with SARS-CoV-2 than among those without SARS-CoV-2 infection and in healthy controls [8,34]. Nonetheless, confirming the impact of the activation of the kynurenine pathway on the pathogenesis, disease progression, and clinical outcomes of COVID-19-related ARDS will require further study.

In previous research, the metabolites of the kynurenine pathway were dysregulated in animal models and in clinical observational studies of acute kidney injury [38]. Elevated plasma kynurenine values and/or the kynurenine/tryptophan ratio are predictive of sepsis and multiple organ failure in patients suffering major trauma [39]. Kynurenine-3-monooxygenase is a key enzyme and drug target involved in the conversion of kynurenine into neuroactive metabolites of the immunoregulatory kynurenine pathway, including 3-hydroxykynurenine, which contribute to increased oxidative stress, cellular damage, and apoptosis through the production of ROS. Efforts to inhibit the release of kynurenine-3-monooxygenase have been shown to reduce 3-hydroxykynurenine levels to alleviate acute kidney injury, prevent multiple organ failure, and ameliorate histological changes in lung tissue that are consistent with ARDS [19,21,38,40]. The IDO-kynurenine-aryl hydrocarbon receptor signaling pathway has also been shown to play an important role in inflammation and multiple organ injuries in cases of SARS-CoV-2 infection [7,9].

The most common cause of death among ARDS patients is multiple organ failure [1,2]. Nonetheless, few reports have explored the links between the kynurenine pathway and multiple organ failure in ARDS. Our findings revealed that plasma kynurenine concentrations were significantly correlated with SOFA scores at the time of ARDS onset, while a Cox regression model revealed that kynurenine levels and the SOFA score were both independently associated with hospital mortality. ARDS is characterized by substantial alveolar and systemic inflammation, triggered by proinflammatory mediators and cytokines [1,2]. Although cytokine analysis was not performed in this study to identify the potential metabolite-cytokine relationships, it is reasonable to assume that in ARDS patients, proinflammatory cytokines seen during ARDS promote IDO activity, thereby contributing to an imbalance between tryptophan and kynurenine, leading to multiple organ failure and death.

This study was hindered by a number of limitations. First, all the patients were from a single tertiary-care referral center, with a small sample size; the cohort had more comorbidities, with higher hospital mortality. Significant numbers of ARDS patients were on neuromuscular blockage medication and, therefore, lacked external validation. This no doubt limits the generalizability and reliability of our findings. Furthermore, we selected unventilated healthy individuals (rather than ventilated ICU patients without ARDS) as our control group. This, no doubt, poses a potential confounder. Second, we did not examine the precise mechanism involved in activating the kynurenine pathway regarding the pathogenesis of ARDS. Third, the measured parameters that reflected inflammation and oxidative stress, such as CRP, were not significantly different between survivors and non-survivors. The circulating cytokines (e.g., IL-6 or interferon-γ) were not measured in this study; further analysis is needed to examine the inflammatory status over the various time points. The markers of oxidative stress were also not examined [41,42]. Therefore, we could not verify whether non-survivors were indeed characterized by increased inflammation and oxidative stress, which was directly related to alterations in the kynurenine pathway. Fourth, despite the fact that the kynurenine/tryptophan ratio tends to fluctuate under certain conditions, we used it as a surrogate for IDO activity (as used in recent studies) and did not check the exact IDO values. Finally, ARDS is a heterogeneous syndrome with complex pathogenesis; the study groups were prone to diverse variations in comorbidities and nutritional status, as well as pharmaceutical and clinical interventions, all of which may have influenced the metabolic fingerprint (metabotypes) of individuals. We observed an association between the activation of the kynurenine pathway and hospital mortality in ARDS patients; however, the causal relationship has yet to be clearly determined since the relationships with endophenotypes (pathways) and the mechanisms of ARDS were not examined.

## 5. Conclusions

In the current prospective metabolic profiling study on patients with ARDS, plasma kynurenine levels and the plasma kynurenine/tryptophan ratio were both independently associated with hospital mortality. Future research should focus on the mechanism(s) underlying kynurenine pathway activation in the pathogenesis of ARDS. It will also be necessary to determine whether the activation of the kynurenine pathway could be considered a biomarker by which to stratify subgroups and predict clinical outcomes. Researchers could also consider therapeutic interventions such as antioxidants, targeting the metabolic dysregulation of the kynurenine pathway to alleviate disease progression and reduce mortality in cases of ARDS.

## Figures and Tables

**Figure 1 antioxidants-11-01884-f001:**
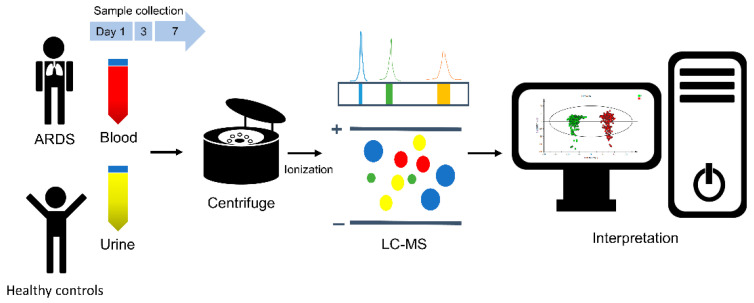
Schematic illustration showing the workflow of the study, involving the metabolomic analysis of biological samples from ARDS patients and healthy controls. Blood and urine samples were collected at day 1, day 3, and day 7 after ARDS onset. Blood and urine samples were also collected from healthy controls. Biological samples were analyzed for their metabolic profiles. ARDS: acute respiratory distress syndrome; LC-MS: liquid chromatography-mass spectrometry.

**Figure 2 antioxidants-11-01884-f002:**
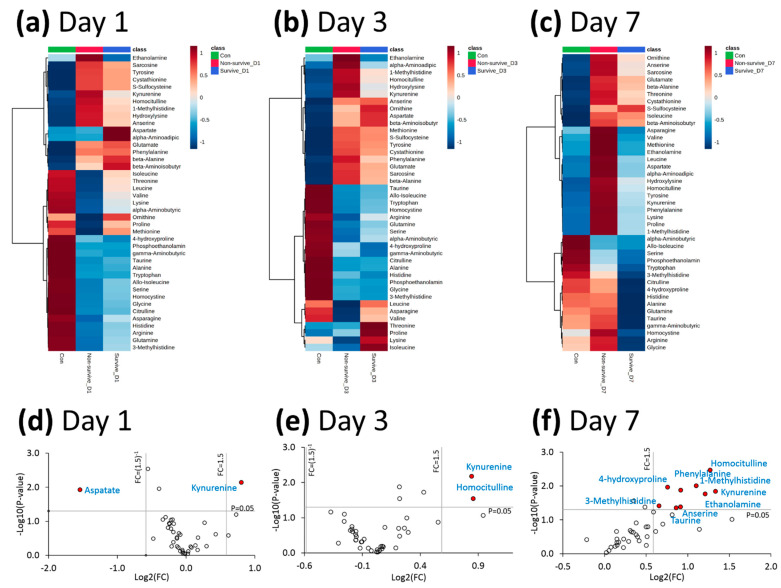
(**a**–**c**) Heat maps of amino acids and biogenic amines among the survivors and non-survivors of ARDS at day 1, day 3, and day 7 after ARDS onset, as well as healthy control subjects. (**d**–**f**) Volcano plots of the amino acids and biogenic amines, comparing survivors and non-survivors of ARDS at day 1, day 3, and day 7 after ARDS onset. Data were selected at the cutoff point for a *p*-value < 0.05 and an FC < 0.66 or >1.5. ARDS: acute respiratory distress syndrome; FC: fold change.

**Figure 3 antioxidants-11-01884-f003:**
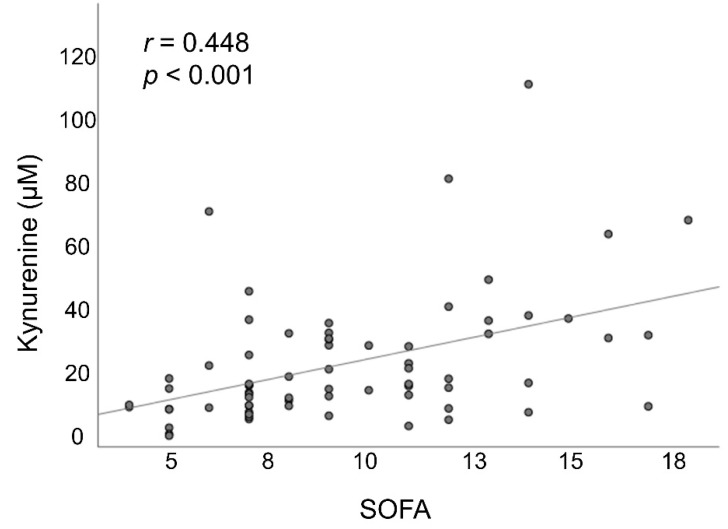
The association between plasma kynurenine levels and SOFA scores at day 1 after ARDS onset. ARDS: acute respiratory distress syndrome; SOFA: sequential organ failure assessment.

**Figure 4 antioxidants-11-01884-f004:**
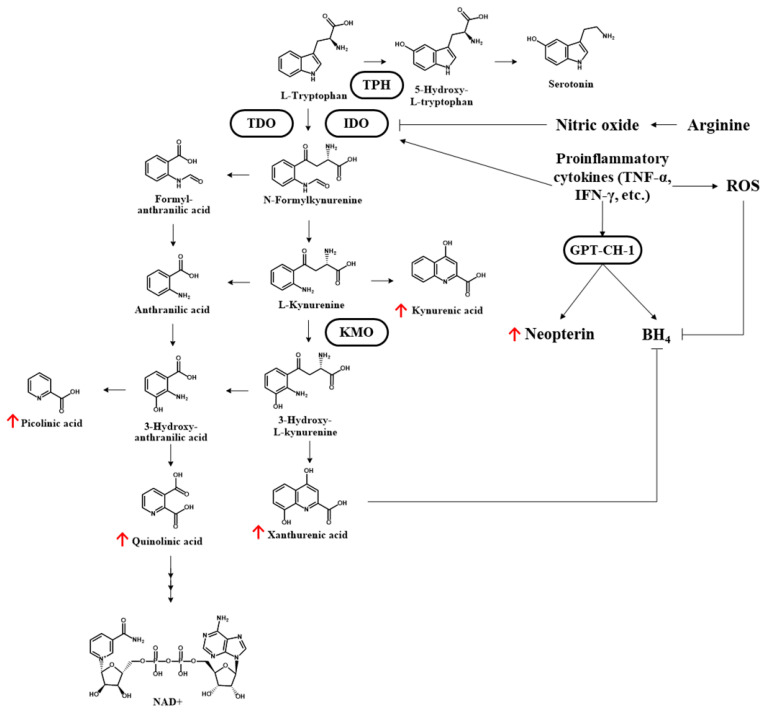
Kynurenine pathway of tryptophan catabolism. The altered metabolites of enrolled ARDS patients are highlighted with red arrows (non-survivors > survivors). TPH: tryptophan 5-hydroxylase; TDO: tryptophan-2,3-dioxygenase; IDO: indoleamine 2,3-dioxygenase; TNF-α: tumor necrosis factor-α; ROS: reactive oxygen species; GTP-CH1: guanosine triphosphate cyclohydrolase 1; KMO: kynurenine-3-monooxygenase; BH4: tetrahydrobiopterin; NAD: nicotinamide adenine dinucleotide.

**Figure 5 antioxidants-11-01884-f005:**
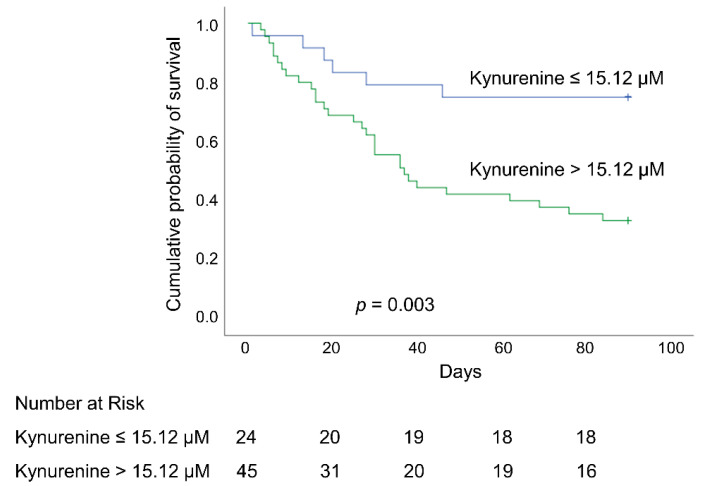
Kaplan–Meier 90-day survival curves of patients with ARDS, as stratified using an optimal cutoff value of plasma kynurenine (15.12 µM) at day 1. ARDS: acute respiratory distress syndrome.

**Table 1 antioxidants-11-01884-t001:** Background characteristics and clinical and metabolic variables: healthy controls and ARDS patients.

Characteristics	Healthy Controls (*n* = 30)	ARDS			*p-*Value
		All (*n* = 69)	Survivors (*n* = 33)	Non-Survivors (*n* = 36)	
Age (years)	66.3 ± 8.9	65.2 ± 14.5	63.9 ± 12.4	66.2 ± 15.9	0.512
Male (gender)	24 (80%)	53 (76.8%)	24 (72.7%)	29 (80.6%)	0.582
Body weight (kg)	68.7 ± 10.1	60.4 ± 11.9	65.2 ± 10.6	56.2 ± 11.1	0.001
Body mass index (kg/m^2^)	24.7 ± 2.9	23.2 ± 4.4	24.8 ± 3.8	21.9 ± 4.4	0.006
ARDS etiologies					
Bacterial pneumonia		50 (72.5%)	22 (66.7%)	28 (77.8%)	0.302
Extrapulmonary sepsis		6 (8.7%)	3 (9.1%)	3 (8.3%)	1.0
Aspiration pneumonia		4 (5.8%)	1 (3%)	3 (8.3%)	0.616
Influenza pneumonia		7 (10.1%)	5 (15.1%)	2 (5.6%)	0.242
Pulmonary hemorrhage		2 (2.9%)	2 (6.1%)	0 (0%)	0.219
Comorbidities					
Hypertension	9 (30%)	24 (34.8%)	8 (24.2%)	16 (44.4%)	0.095
Diabetes mellitus	3 (10%)	22 (31.9%)	11 (33.3%)	11 (30.6%)	0.746
Chronic lung disease		14 (20.3%)	5 (15.2%)	9 (25%)	0.338
Chronic liver disease		8 (11.6%)	5 (15.2%)	3 (8.3%)	0.462
Chronic kidney disease		20 (29%)	6 (18.2%)	14 (38.9%)	0.069
Immunocompromised status		32 (46.4%)	10 (30.3%)	22 (61.1%)	0.015
SOFA score at day 1		9.5 ± 3.5	7.9 ± 2.8	10.9 ± 3.5	<0.001
Lung injury score at day 1		2.91 ± 0.46	2.84 ± 0.46	2.98 ± 0.43	0.205
Metabolic profiles					
Kynurenine at day 1 (µM)	3.7 ± 0.6	25.1 ± 19.1	17.8 ± 10.4	31.7 ± 22.6	0.002
Kynurenine at day 3 (µM)		29.8 ± 25.2	19.4 ± 11.1	39.5 ± 30.6	0.001
Kynurenine at day 7 (µM)		29.3 ± 39.6	16.8 ± 11.0	42.2 ± 52.7	0.014
Kynurenine/tryptophan ratio at day 1		1.1 ± 0.8	0.8 ± 0.6	1.4 ± 0.9	0.002
Kynurenine/tryptophan ratio at day 3		1.2 ± 1.2	0.7 ± 0.5	1.6 ± 1.5	0.003
Kynurenine/tryptophan ratio at day 7		0.8 ± 1.0	0.5 ± 0.3	1.2 ± 1.3	0.013
Laboratory data at day 1					
WBC (10^3^/µL)	5.9 ± 1.6	12.3 ± 6.2	11 ± 5.1	13.4 ± 7	0.115
Segment (%)		82 ± 14	81 ± 14	83 ± 13	0.649
Band (%)		2 (0–5)	1 (0–4)	3 (1–5)	0.493
Lactate (mg/dL)		15.3 (10.2–20.9)	14 (9.7–17.7)	17.7 (11–22.1)	0.057
CRP (mg/L)		161 (109–263)	146 (101–248)	172 (115–261)	0.406
Procalcitonin (ng/mL)		6.3 (1.7–27)	5.4 (0.9–19.1)	6.9 (2.5–29.1)	0.272
PaO_2_/FiO_2_ (mm Hg) at day 1		164.5 ± 62.2	177.9 ± 64	152.5 ± 56.9	0.087
FiO_2_ (%) at day 1		60.4 ± 18.2	59.8 ± 18.6	60.9 ± 17.5	0.803
Ventilator settings at day 1					
Mechanical power (J/min)		19.7 ± 6.1	18.5 ± 4.8	20.8 ± 6.9	0.118
Tidal volume (ml/kg PBW)		8.5 ± 2.4	8.1 ± 1.8	8.8 ± 2.7	0.212
PEEP (cm H_2_O)		11.0 ± 1.9	11.2 ± 2.1	10.9 ± 1.7	0.576
Peak inspiratory pressure (cm H_2_O)		28.1 ± 5.4	27.3 ± 4.8	28.8 ± 5.6	0.253
Mean airway pressure (cm H_2_O)		16.5 ± 3.0	16.4 ± 3.0	16.7 ± 2.9	0.662
Dynamic compliance (ml/cm H_2_O)		30.0 ± 12.9	30.7 ± 12.1	29.4 ± 13.4	0.676
Total respiratory rate (breaths/min)		22.2 ± 4.2	21.8 ± 4.1	22.6 ± 4.3	0.445
SOFA score at day 3		9.5 ± 4.1	7.8 ± 3.3	11.0 ± 4.0	<0.001
PaO_2_/FiO_2_ (mm Hg) at day 3		173.7 ± 70.2	191.6 ± 75.6	157.7 ± 59.7	0.024
FiO_2_ (%) at day 3		52.4 ± 17.6	47.3 ± 15.3	56.9 ± 18.3	0.026
SOFA score at day 7		8.8 ± 5.0	5.7 ± 3.2	11.9 ± 4.5	<0.001
PaO_2_/FiO_2_ (mm Hg) at day 7		178.6 ± 89.5	213.2 ± 90.5	151.5 ± 75.8	0.004
FiO_2_ (%) at day 7		53.0 ± 20.2	45.7 ± 13.5	59.2 ± 22.7	0.006
Duration of mechanical ventilation (days)		17 (7–30)	9 (7–21)	20 (11–34)	0.075
Length of ICU stay (days)		18 (9–37)	11 (9–31)	20 (11–38)	0.5
Length of hospital stay (days)		32 (19–56)	40 (21–60)	30 (13–47)	0.135
28-day survival time (days)		23.0 ± 8.7	28.0 ± 0	18.5 ± 10.1	<0.001
60-day survival time (days)		41.3 ± 21.9	60.0 ± 0	24.7 ± 17.9	<0.001
90-day survival time (days)		56.2 ± 35.6	90.0 ± 0	26.1 ± 21.1	<0.001

Values are expressed as *n* (%), mean ± standard deviation, or median (25th–75th percentiles). ARDS: acute respiratory distress syndrome; CRP: C-reactive protein; FiO_2_: fraction of inspired oxygen; ICU: intensive care unit; PaO_2_: partial pressure of oxygen in arterial blood; PBW: predicted body weight; PEEP: positive end-expiratory pressure; SOFA: sequential organ failure assessment; WBC: white blood cells.

**Table 2 antioxidants-11-01884-t002:** The background characteristics, clinical variables, and outcomes as a function of plasma kynurenine at ARDS onset.

Characteristics	Kynurenine at ARDS Onset		*p*-Value
	High (*n* = 45) (>15.12 µM)	Low (*n* = 24) (≤15.12 µM)	
Age (years)	66.1 ± 15.2	62.4 ± 12.3	0.310
Male (gender)	37 (82.2%)	16 (66.7%)	0.145
Body weight (kg)	61.4 ± 10.9	59.7 ± 12.4	0.563
Body mass index (kg/m^2^)	23.6 ± 4.1	22.9 ± 4.6	0.516
SOFA score at day 1	10.3 ± 3.3	7.8 ± 3.3	0.004
SOFA score at day 3	10.8 ± 3.9	7.0 ± 3.3	<0.001
SOFA score at day 7	10.4 ± 5.0	5.8 ± 3.2	<0.001
Lung injury score	2.9 ± 0.4	2.9 ± 0.5	0.717
Metabolic profiles			
Kynurenine at day 1 (µM)	32.9 ± 19.4	10.4 ± 3.1	<0.001
Kynurenine at day 3 (µM)	38.2 ± 27.1	13.2 ± 6.4	<0.001
Kynurenine at day 7 (µM)	38.5 ± 46.9	12.9 ± 7.4	0.002
Kynurenine/tryptophan ratio at day 1	1.4 ± 0.9	0.5 ± 0.2	<0.001
Kynurenine/tryptophan ratio at day 3	1.5 ± 1.3	0.5 ± 0.3	<0.001
Kynurenine/tryptophan ratio at day 7	1.1 ± 1.2	0.4 ± 0.2	0.001
Laboratory data at day 1			
WBC (×10^3^/µL)	12.9 ± 6.6	11.3 ± 5.5	0.333
Segment (%)	82 ± 14	81 ± 13	0.678
Band (%)	3 (1–5)	1 (0–5)	0.636
Lactate (mg/dL)	15.1 (10.6–21.1)	15.6 (9.9–19.7)	0.698
CRP (mg/L)	158 (110–260)	171 (76–303)	0.850
Procalcitonin (ng/mL)	6.9 (2.4–27.7)	4.0 (0.4–18.3)	0.072
PaO_2_/FiO_2_ (mm Hg) at day 1	162.6 ± 61.3	172.4 ± 62.3	0.533
Ventilator settings at day 1			
Mechanical power (J/min)	19.9 ± 6.6	19.6 ± 5.1	0.825
Tidal volume (ml/kg PBW)	8.3 ± 2.7	8.6 ± 1.6	0.681
PEEP (cm H_2_O)	11.2 ± 1.9	10.9 ± 2.0	0.574
Peak inspiratory pressure (cm H_2_O)	28.6 ± 5.2	27.7 ± 5.4	0.497
Mean airway pressure (cm H_2_O)	16.8 ± 2.8	16.2 ± 3.3	0.444
Dynamic compliance (ml/cm H_2_O)	28.8 ± 12.6	31.3 ± 12.8	0.438
Total respiratory rate (breaths/min)	22.4 ± 4.3	21.9 ± 4.2	0.644
Outcomes			
Duration of mechanical ventilation (days)	17.5 (8–34)	9 (7–20)	0.204
Length of ICU stay (days)	20 (9–38)	13.5 (8.3–33.8)	0.650
Length of hospital stay (days)	34 (20–59)	25 (16–53.3)	0.451
28-day survival time (days)	22.4 ± 8.8	25.1 ± 7.3	0.180
60-day survival time (days)	37.5 ± 21.9	50.3 ± 18.6	0.018
90-day survival time (days)	48.6 ± 34.5	72.8 ± 31.3	0.006
Hospital mortality	30 (66.7%)	6 (25%)	0.003

Values are presented as *n* (%), mean ± standard deviation, or median (25th–75th percentiles). ARDS: acute respiratory distress syndrome; CRP: C-reactive protein; FiO_2_: fraction of inspired oxygen; ICU: intensive care unit; PaO_2_: partial pressure of oxygen in arterial blood; PBW: predicted body weight; PEEP: positive end-expiratory pressure; SOFA: sequential organ failure assessment; WBC: white blood cells.

**Table 3 antioxidants-11-01884-t003:** Urine metabolites of tryptophan degradation at day 1, day 3, and day 7 after ARDS onset, compared to survivors at day 1.

Metabolites	Day 1		Day 3			Day 7		
	Non-Survivors	*p-*Value	Survivors	Non-Survivors	*p-*Value	Survivors	Non-Survivors	*p-*Value
	Fold Change		Fold Change	Fold Change		Fold Change	Fold Change	
Picolinic acid	1.52	0.002	1.22	1.43	<0.001	1.45	1.79	0.139
Neopterin	1.87	0.003	1.09	1.67	0.044	1.06	2.85	0.003
Nicotinic acid or Quinolinic acid	1.47	<0.001	1.11	1.69	<0.001	1.21	2.00	0.017
3-Hydroxykynurenine	1.30	0.299	1.05	1.57	0.055	0.74	0.95	0.192
Kynurenine	1.19	0.392	1.33	1.86	0.062	1.44	1.20	0.478
3-Hydroxyanthranilic acid	1.20	0.383	1.15	1.39	0.215	2.30	2.31	0.982
Tryptophan	0.72	0.031	1.08	0.95	0.282	1.07	1.10	0.850
Xanthurenic acid	1.43	0.035	0.96	1.49	0.010	1.27	5.62	0.038
Kynurenic acid	1.97	<0.001	0.99	1.84	<0.001	1.21	2.81	0.003

Data were calculated by dividing the value of metabolites by the value of metabolites of the survivors at day 1. ARDS: acute respiratory distress syndrome.

**Table 4 antioxidants-11-01884-t004:** Cox proportional hazard regression analysis of factors associated with hospital mortality.

Variables	Univariate Analysis		Multivariable Analysis Model 1		Multivariable Analysis Model 2		Multivariable Analysis Model 3	
	HR (95% CI)	*p-*Value	Adjusted HR (95% CI)	*p*-Value	Adjusted HR (95% CI)	*p-*Value	Adjusted HR (95% CI)	*p-*Value
Age	1.010 (0.986–1.035)	0.414						
Body mass index	0.896 (0.827–0.971)	0.007						
Bacterial pneumonia	1.468 (0.670–3.213)	0.337						
Influenza pneumonia	0.463 (0.111–1.927)	0.290						
Hypertension	1.738 (0.906–3.335)	0.096						
Chronic kidney disease	1.880 (0.965–3.663)	0.064						
Immunocompromised status	2.073 (1.072–4.008)	0.030	3.284 (1.559–6.917)	0.002	2.730 (1.313–5.676)	0.007	2.963 (1.434–6.121)	0.003
SOFA score at day 1	1.218 (1.110–1.336)	<0.001	1.136 (1.017–1.269)	0.024	1.173 (1.055–1.305)	0.003	1.130 (1.009–1.266)	0.034
Lung injury score at day 1	1.740 (0.865–3.501)	0.120						
WBC at day 1	1.000 (1.000–1.000)	0.201						
Lactate at day 1	1.024 (1.010–1.038)	0.001	1.033 (1.012–1.054)	0.002	1.018 (1.004–1.032)	0.014	1.032 (1.011–1.053)	0.003
Procalcitonin at day 1	1.003 (0.997–1.008)	0.346						
PaO_2_/FiO_2_ at day 1	0.994 (0.988–0.999)	0.030						
MP at day 1	1.034 (0.985–1.086)	0.177						
V_T_/PBW at day 1	1.097 (0.961–1.251)	0.170						
PIP at day 1	1.026 (0.969–1.086)	0.378						
Total RR at day 1	1.031 (0.956–1.112)	0.428						
Kynurenine at day 1	1.018 (1.006–1.032)	0.005	1.017 (1.003–1.032)	0.017				
Kynurenine/tryptophan ratio at day 1	1.785 (1.295–2.461)	<0.001			1.761 (1.204–2.576)	0.004		
Kynurenine at day 1 > 15.12 µM	3.806 (1.579–9.179)	0.003					4.317 (1.621–11.495)	0.003

CI: confidence interval; FiO_2_: fraction of inspired oxygen; HR: hazard ratio; MP: mechanical power; PaO_2_: partial pressure of oxygen in arterial blood; PBW: predicted body weight; PIP: peak inspiratory pressure; RR: respiratory rate; SOFA: sequential organ failure assessment; V_T_: tidal volume; WBC: white blood cells.

## Data Availability

All of the data is contained within the article and the Appendix A.

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
