# Peer review of "Kynurenine Pathway of Tryptophan Metabolism Is Associated with Hospital Mortality in Patients with Acute Respiratory Distress Syndrome: A Prospective Cohort Study"

_antioxidants, 2022, doi:10.3390/antiox11101884_

Round 1
Reviewer 1 Report
Major:
Fig 5. If this fig is adapted from an other article, please correct the title and indicate the reference used for generating the figure.
Author Response
Point 1: Major:
Fig 5. If this fig is adapted from an other article, please correct the title and indicate the reference used for generating the figure.
Response 1: We thank the reviewer’s question and comment. All figures including Figure 5 were depicted by the authors themselves and was not adapted from an other article.
We thank the reviewer for valuable comments. Addressing them fully has significantly strengthened the manuscript.
Reviewer 2 Report
The authors conducted a prospective study into the relation between plasma amino acid and biogenic amine levels and ARDS mortality.
In the abstract and in the introduction, the study is broadly described as a metabolomics study. It should be made more clear it is not a broad OMICS study, but specific to amino acid and amines. As such the objective should be rephrased. Moreover, the introduction should explain better why this was the focus, this is now present only in the discussion. In addition, the impression is created that the study will examine relations to endo/phenotypes (pathways) of ARDS, which it does not.
A recent paper by Metwaly in AJP (2021) on metabolomics in ARDS should be included as it also finds increased kynurenine levels that remain high in ARDS patients. Importantly, this study did include better control groups.
Although data are present on all measured amino acids, they are not mentioned. At all timepoints other amino acids seem altered between controls and ARDS patients, between survivors and non-survivors. It should be specified which ones are statistically different, at least in the supplemental table. It seems true that only kynurenine is consistently different over all time points, but this cannot be judged from the current presentation of the results. It should also be commented on why the inverse is not observed for tryptophane and why the ratio is not as elevated as was reported in previous studies according to your discussion.
The results should start with a description of the patient and control characteristics and not with data. In the methods in/exclusion criteria for controls are not mentioned, nor how these subjects were recruited.
It is not clear why the urine data are not used to assess relations to mortality. Moreover, it would be good to include the survival time and cause of death. It seems like the overall 90-day survival is not the same as the in hospital survival. Please clarify or correct accordingly to use the same nomenclature.
In the discussion the authors make bold statements that inflammation is responsible for the increased metabolism of tryptophane. Yet, there are no data to support this claim from the current study. In contrast, there is no difference in CRP between survivors and non-survivors. It is mentioned in line 346-347 as a limitation that OUR measurements of cytokines were not integrated into the metabolomics data. This suggests that data are available and should be included. If not available, such analyses should be performed to support the claims. However, it is also confusing that inflammation is a trigger to IDO transcription while the metabolites would have anti-inflammatory effects. Overall, although the authors show a relation of kynurenine levels to survival after ARDS, it is entirely unclear what mechanism could be involved. At least some more effort should be put into examining the inflammatory status over the various timepoints to also receive some insight into cause and effect.
The explanation as to how patients were divided into high vs low kynurenine levels comes to early. The subgroups are reported only in table 2, where the explanation should be given.
Author Response
The authors conducted a prospective study into the relation between plasma amino acid and biogenic amine levels and ARDS mortality.
Point 1: In the abstract and in the introduction, the study is broadly described as a metabolomics study. It should be made more clear it is not a broad OMICS study, but specific to amino acid and amines. As such the objective should be rephrased. Moreover, the introduction should explain better why this was the focus, this is now present only in the discussion. In addition, the impression is created that the study will examine relations to endo/phenotypes (pathways) of ARDS, which it does not.
Response 1: We thank the reviewer’s comment to point out this problem. We agreed with the reviewer that this study is not a broad OMICS study, but specific to metabolic profiling of amino acids and biogenic amines. We rephrased the descriptions in the the abstract and the introduction, and revised “metabolomics study” as “study on metabolic profiling specific to amino acids and biogenic amines” in the revised manuscript.
This prospective study on metabolic profiling of amino acids found that kynurenine accumulation was associated with mortality of patients with ARDS. However, the exact endophenotypes (pathways) were not explored and required further investigation in the future. Our currently ongoing ARDS research projects about metabolic profiling study will perform cell experiments to investigate the pathways and mechanisms.
We explained why amino acids and biogenic amines was the focus in the fourth paragraph of Introduction section in the revised manuscript as follows (marked with red text):
The extensive ROS production contributes to oxidative stress can increase the permeability of the pulmonary endothelial bed and plays an important role in the pathogenesis and progression in ARDS. The activation of kynurenine pathway was linked with an increased oxidative stress, and kynurenine can exert immunosuppressive and anti-inflammatory effects via the ROS pathway.
We addressed the limitations in the eighth paragraph of Discussion section in the revised manuscript as follows (marked with red text):
We observed an association between the activation of the kynurenine pathway and hospital mortality in ARDS patients; however, the causal relationship has yet to be clearly determined, and the relations to endophenotypes (pathways) and mechanisms of ARDS were not examined.
Point 2: A recent paper by Metwaly in AJP (2021) on metabolomics in ARDS should be included as it also finds increased kynurenine levels that remain high in ARDS patients. Importantly, this study did include better control groups.
Response 2: We thank the reviewer’s suggestion to include the recent paper by Metwaly in AJP (2021), which did recruit better control groups (i.e., ICU ventilated controls). They found higher kynurenine levels in ARDS patients than ICU ventilated controls at day 1, day 7, day 14, and day 28 after ICU admission. We cited this paper as Reference 30 in the revised manuscript.
We added the descrptions in the second paragraph of Discussion section in the revised manuscript as follows (marked with red text):
A recent study reported that higher kynurenine levels were found in ARDS patients than in ICU ventilated controls at day 1, day 7, day 14, and day 28 after ICU admission.
Reference:
- Metwaly, S.; Côté, A.; Donnelly, S.J.; Banoei, M.M.; Lee, C.H.; Andonegui, G.; Yipp, B.G.; Vogel, H.J.; Fiehn, O.; Winston, B.W. ARDS metabolic fingerprints: characterization, benchmarking, and potential mechanistic interpretation. Am J Physiol Lung Cell Mol Physiol. 2021, 321, L79–L90.
Point 3: Although data are present on all measured amino acids, they are not mentioned. At all timepoints other amino acids seem altered between controls and ARDS patients, between survivors and non-survivors. It should be specified which ones are statistically different, at least in the supplemental table. It seems true that only kynurenine is consistently different over all time points, but this cannot be judged from the current presentation of the results. It should also be commented on why the inverse is not observed for tryptophane and why the ratio is not as elevated as was reported in previous studies according to your discussion.
Response 3: We appreciated the reviewer’s comments and apologized for the lack of detailed reports of all measured amino acids and biogenic amines among healthy controls and ARDS patients.
We provided all measured amino acids and biogenic amines of healthy controls and between survivors and nonsurvivors (day 1 to day 7 after ARDS onset) in the Supplementray Materials (Tables S1 and S2). The differences of metabolites were compared between survivors and non-survivors at day 1, day 3 and day 7 after onset of ARDS, and p value were shown.
Tryptophan is an essential amino acid and obtained entirely via dietary intake. Tryptophan is metabolized for protein biosynthesis, and through kynurenine or serotonin (5-hydroxytryptamine, 5-HT) pathways [1, 2]. We speculated that there may be variations in the dietary intakes and tryptophan metabolisms between survivors and nonsurvivors of ARDS patients, and we did not explore the tryptophan metabolic pathways (i.e., kynurenine pathway and serotonin pathway) extensively, which may cause the inverse is not observed for tryptophan.
The kynurenine/tryptophan ratio often indicates the IDO activity. However, this ratio may depend on the metabolism and the half-life of kynurenine and tryptophan. Although biological samples were obtained from ARDS patients after fasting overnight (for at least 8 hours), administration of tryptophan-containing parenteral infusions were varying among ARDS patients which could also influence the results.
However, nonsurvivors of ARDS patients experienced severe inflammation than survivors with higher oxidative stress, which contributed to kynurenine accumulations in our study.
We revised Table S1 and Table S2 in the Supplementray Materials.
We added the comments to explain why the inverse is not observed for tryptophan and why the ratio is not as elevated as was reported in previous studies in the fourth paragraph of Discussion section as follows (marked with red text):
Higher kynurenine levels were found in ARDS patients, but the inverse is not observed for tryptophan in this study. The kynurenine/tryptophan ratio in this study is not as elevated as previous reported studies. It may be related to the half-life of kynurenine and tryptophan, variations in tryptophan-containing parenteral infusions and tryptophan metabolism pathways (protein biosynthesis, serotonin and kynurenine pathways) among patients with ARDS.
References:
- Platten, M.; Nollen, E.A.A; Röhrig, U.F.; Fallarino, F.; Opitz, C.A. Tryptophan metabolism as a common therapeutic target in cancer, neurodegeneration and beyond. Nat Rev Drug Discov. 2019, 18, 379–
- Cervenka, I.; Agudelo, L.Z.; Ruas, J.L. Kynurenines: Tryptophan's metabolites in exercise, inflammation, and mental health. Science. 2017, 357, eaaf9794.
Point 4: The results should start with a description of the patient and control characteristics and not with data. In the methods in/exclusion criteria for controls are not mentioned, nor how these subjects were recruited.
Response 4: We thank the reviewer’s suggestions.
We revised the data of healthy controls in Table 1 and described the characteristics of ARDS patients and healthy controls in the first paragraph of the Results section (3.1. Study Patients) as follows (marked with red text):
The mean age of healthy control and ARDS patients were 66.3 and 65.2 years (p = 0.532). Twenty-four persons (80%) and 53 patients (76.8%) were male in healthy controls and ARDS (p = 0.726) . The main cause of ARDS was bacterial pneumonia (n = 50, 72.5%), followed by influenza pneumonia (n = 7, 10.1%). There were no significant comorbidities among healthy controls, 9 persons (30%) had hypertension and 3 persons (10%) had diabetes mellitus. Immunocompromised status (32 patietns, 46.4%) was the most common comorbidity in ARDS patients, followed by hypertension (24 patients, 34.8%). Thirty-six patients with ARDS died, and overall hospital mortality was 52.2%.
We added the inclusion and exclusion criteria for healthy controls in the first paragraph of Materials and Methods section (2.1. Study Design and Patients Cohort) as follows (marked with red text):
Inclusion criteria for healthy control subjects were sex- and age-matched residents who were living at Chang Gung Health and Culture Village in Taiwan. Exclusion criteria for healthy controls were persons who had evidences of significant comorbidities.
Point 5: It is not clear why the urine data are not used to assess relations to mortality. Moreover, it would be good to include the survival time and cause of death. It seems like the overall 90-day survival is not the same as the in hospital survival. Please clarify or correct accordingly to use the same nomenclature.
Response 5: We thank the reviewer’s comments and suggestions. The most common cause of death in ARDS is multiple organ failure [1]. Many ARDS patients may experience acute kidney injury with elevating creatinine, lower urine output, oliguria or even anuria. Twenty patients (29%) had comorbitidy of chronic kidney disease with poor renal function in our study.
The concentration of each urine metabolite was normalized to each urine sample’s corresponding creatinine value to compensate for urine volumes variations (i.e., the concentration of urine metabolites is presented as (µM /µM creatinine) [2]. However, some patients with ARDS had acute kidney injury and underlying chronic kidney diseases with elevated creatinine and low urine output (or even oliguria), which may make statistics not practical than serum metabolites and therefore limited the numbers of patients in this prospective study for statistical analysis.
Although it was reasonable to assume that serum and urine metabolites were correlated well to predict clinical outcomes, we appreciated and agreed with the reviewer’s comment that urine metabolites could be better predictors for outcomes in ARDS patients. We will recruit more ARDS patients and integrate blood and urine metabolites to develop valuable biomarkers in our ongoing research projects in ARDS.
We added the 28-day, 60-day and 90-day survival time at Table 1 and Table 2 in the revised manuscript.
Cause of death was provied in the first paragraph of the Results section (3.1. Study Patients) as follows (marked with red text):
The most common cause of death was sepsis with multiple organ failure (n = 24, 66.7%), followed by respiratory failure (n = 7, 19.4%), underlying immunocompromised status (n = 4, 11.1%), and profound cardiogenic shock (n = 1, 2.8%).
We agreed with the reviewer that the overall 90-day survival is not the same as the in hospital survival and apologized for the lack of clarity for the same nomenclature. We used the same nomenclature as “hospital mortality” in the revised manuscript.
References:
- Thompson, B.T.; Chambers, R.C.; Liu, K.D. Acute Respiratory Distress Syndrome. N. J. Med. 2017, 377, 562–572.
- Bouatra, S.; Aziat, F.; Mandal, R.; Guo, A.C.; Wilson, M.R.; Knox, C.; Bjorndahl, T.C.; Krishnamurthy, R.; Saleem, F.; Liu, P.; et al. The human urine metabolome. PLoS One. 2013, 8, e73076.
Point 6: In the discussion the authors make bold statements that inflammation is responsible for the increased metabolism of tryptophane. Yet, there are no data to support this claim from the current study. In contrast, there is no difference in CRP between survivors and non-survivors. It is mentioned in line 346-347 as a limitation that OUR measurements of cytokines were not integrated into the metabolomics data. This suggests that data are available and should be included. If not available, such analyses should be performed to support the claims. However, it is also confusing that inflammation is a trigger to IDO transcription while the metabolites would have anti-inflammatory effects. Overall, although the authors show a relation of kynurenine levels to survival after ARDS, it is entirely unclear what mechanism could be involved. At least some more effort should be put into examining the inflammatory status over the various timepoints to also receive some insight into cause and effect.
Response 6: We appreciate the reviewer’s comments to point out several issues and suggestions, and this is an excellent point of view.
Pneumonia, aspiration pneumonia and sepsis account for 85 % of ARDS [1]. In our study, pneumonia (bacterial, aspiration, and influenza) and sepsis, total 67 patients (97.1%), also account for most causes of ARDS. The value of CRP was higher in nonsurvivors than in survivors at day 1 of ARDS onset ( 172 (115–261) vs. 146 (101–248) , p = 0.406), although the value did not reach significance.
Besides, WBC count, segment form, band form, and procalcitonin were all higher in nonsurvivors than in survivors although these values also did not reach significant difference. These findings may indicate that nonsurvivors had severe inflammatory response than survivors.
Patients with ARDS could suffer from severe oxidative stress from neutrophil activation and high inspired oxygen concentrations (FiO2) used [2]. Neutrophils (the predominant inflammatory cell in ARDS) recruited to the site of lung inflammation generate ROS production causing high oxidative stress [3–5]. Oxidative stress is known to activate multiple intracellular signaling, which induces apoptosis or cell overgrowth, contributing to increased endothelial permeability and eventually multiple organ dysfunction. Therefore, severe inflammation leading to higher oxidative stress in ARDS.
We revised the Table 1 and provided PaO2/FiO2, FiO2, and SOFA scores over the timepoints after ARDS onset (day 1, day 3, and day 7). The results showed that nonsurvivors had significantly lower PaO2/FiO2 and higher FiO2 than survivors at day 3 and day 7 after ARDS onset (all p < 0.05). It indicated that nonsurvivors had higher oxidative stress (i.e., under higher FiO2 treatment) than survivors, which may be arised from severe pulmonary or systemic inflammations. The most common cause of death in ARDS is multiple organ failure [1], and nonsurvivor also had significantly higher SOFA than survivors at day 1, day 3, and day 7 after ARDS onset.
We apologized that we did not check cytokines during this prospective study (2017/02–2018/06), and needed further analysis to be performed. We will perform cytokines analysis in the currently ongoing ARDS research project about metabolomics to support the claims.
However, the above findings of this study indicated that nonsurvivors of ARDS patients experienced more severe inflammation than survivors (higher WBC count, segment form, band form, and procalcitonin), which led to higher oxidative stress (i.e., needed higher FiO2 treatment) and organ failure (i.e., higher SOFA score). Besides, nonsurvivors had higher lactate levels than survivors at day 1 of ARDS onset (14 (9.7–17.7) vs. 17.7 (11–22.1), p= 0.057).
We agreed with the reviewer that inflammation is a trigger to IDO transcription while the metabolites would have anti-inflammatory effects. Both proinflammatory and anti-inflammatory cytokines or mediators could exert various effects on IDO activity. In other words, the IDO status was determined by the balance between proinflammatory and anti-inflammatory cytokines or mediators. Tryptophan catabolites are absorbed through the intestinal epithelium and enter the bloodstream, and some metabolites have anti-oxidative and anti-inflammatory effects. The metabolites of kynurenine pathway largely had immunosuppressive and anti-inflammatory effects [6, 7]. Kynurenine has known anti-inflammatory effects that suppresses the proliferation of effector T cells and induce cell death by apoptosis [7, 8]. Our study didn’t explore the tryptophan metabolic pathways (i.e., kynurenine pathway and serotonin pathway) extensively to evaluate the imbalance between inflammatory and anti-inflammatory effects.
We observed an association between the activation of the kynurenine pathway and mortality in ARDS patients. However, the exact mechanism was not investigated in this study. We agreed with the reviewer that further research was needed to explore the mechanism and inflammatory status over the various timepoints with related metabolic dysregulations, and examine the causal relationship. We will verify the exact mechanisms in our currently ongoing ARDS research projects.
We added the statements that metabolites would have anti-inflammatory effects in the second paragraph of Discussion section.
We addressed the above limitations in the eighth paragraph of Discussion section in the revised manuscript as follows (marked with red text):
Third, our measurements of circulating cytokines (e.g., IL-6 or interferon-γ) were not checked and integrated with metabolic profiling to identify the potential metabolite-cytokine relationships, and needed further analysis to examine the inflammatory status over the various timepoints.
We observed an association between the activation of the kynurenine pathway and hospital mortality in ARDS patients; however, the causal relationship has yet to be clearly determined, and the relations to endophenotypes (pathways) and mechanisms of ARDS were not examined.
References:
- Thompson, B.T.; Chambers, R.C.; Liu, K.D. Acute Respiratory Distress Syndrome. N. J. Med. 2017, 377, 562–572.
- Quinlan, G.J.; Lamb, N.J.; Tilley, R.; Evans, T.W.; Gutteridge, J.M. Plasma hypoxanthine levels in ARDS: implications for oxidative stress, morbidity, and mortality. Am J Respir Crit Care Med. 1997, 155, 479–
- Chow, C.W.; Herrera, Abreu. M.T; Suzuki, T.; Downey, G.P. Oxidative stress and acute lung injury. Am J Respir Cell Mol Biol. 2003, 29, 427–
- Wang,; Liu, D.; Song, P.; Zou, M.H. Tryptophan-kynurenine pathway is dysregulated in inflammation, and immune activation. Front Biosci (Landmark Ed). 2015, 20, 1116–43.
- Kellner, M.; Noonepalle, S.; Lu, Q.; Srivastava, A.; Zemskov, E.; Black, S.M. ROS Signaling in the Pathogenesis of Acute Lung Injury (ALI) and Acute Respiratory Distress Syndrome (ARDS). Adv Exp Med Biol. 2017, 967, 105–
- Haq, S.; Grondin, J.A.; Khan, W.I. Tryptophan-derived serotonin-kynurenine balance in immune activation and intestinal inflammation. FASEB J. 2021, 35, e21888.
- Cervenka, I.; Agudelo, L.Z.; Ruas, J.L. Kynurenines: Tryptophan's metabolites in exercise, inflammation, and mental health. Science. 2017, 357, eaaf9794.
- Roager, H.M.; Licht, T.R. Microbial tryptophan catabolites in health and disease. Nat Commun. 2018, 9, 3294.
Point 7: The explanation as to how patients were divided into high vs low kynurenine levels comes to early. The subgroups are reported only in table 2, where the explanation should be given.
Response 7: We thank the reviewer’s comment and suggestion. Patients with ARDS sufferred from high mortality. Mild, moderate, and severe ARDS were associated with increased mortality (27%, 32%, and 45%, respectively) [1], and therefore some patients with ARDS may die within one week or during the nitial phase of ARDS. Therefore, early identification of the biological or clinical variables (like at day 1 of ARDS onset, rather than at day 7) to predict disease progression or mortality in ARDS is crucial.
The predictive values of kynurenine on hospital mortality in our study were as follows: the area under the receiver operating characteristic (AUROC) of kynurenine at day 1, day 3 and day 7 were 0.73 (p = 0.002), 0.687 (p = 0.008), and 0.73 (p = 0.002), respectively. It indicated that the predictive value of kynurenine at day 1 was not inferior to kynurenine at day 3. Therefore, we categorize patients according to plasma kynurenine levels at day 1.
We explained the above descriptions in the second paragraph of Results section (3.1. Study Patients) in the revised manuscript as follows (marked with red text):
For early identification of biomarkers to predict outcomes of patients with ARDS, the maximum Youden index value was used to categorize patients according to plasma kynurenine levels at day 1, using a cutoff of 15.12 µM: high plasma kynurenine group (45 patients; 65 %) and low plasma kynurenine group (24 patients; 35 %).
Refererences:
- ARDS Definition Task Force, Ranieri, V.M.; Rubenfeld, G.D.; Thompson, B.T.; Ferguson, D.; Caldwell, E.; Fan, E.; Camporota, L.; Slutsky, A.S. Acute respiratory distress syndrome: the Berlin Definition. JAMA. 2012, 307, 2526–2533.
We thank the reviewer for valuable comments. Addressing them fully has significantly strengthened the manuscript.
Reviewer 3 Report
The manuscript by Chiu et al. focuses on evaluating multiple metabolites in serum and urine from patients with ARDS. Information from this study may provide new biomarkers or targets for intervention, as the authors suggested that kynurenine is associated with mortality. The following is a synopsis of concerns raised with the data presented in the manuscript.
Major Concerns:
1) It is unclear how the authors define “mortality.” Does the mortality have a time limit, such as 28-day or 90-day mortality?
2) What are the benefits of measuring kynurenine compared to existing models, such as the SOFA score?
3) Significant non-survivals have CKD and are immunocompromised hosts.
4) Kynurenine levels correlate well, mostly on day 1, but not in the later time points (3 and 7 days).
5) It is unclear whether kynurenine is truly an antioxidant, or whether this manuscript fits the overall theme of the journal. The article might be more suitable for a journal such as Critical Care Medicine, if the authors can justify the pitfalls of their study design.
6) Significant numbers of ARDS patients were on neuromuscular blockage, a practice less commonly used in the U.S. This raises the question if the data gathered in the study could be applied to a general environment.
Author Response
The manuscript by Chiu et al. focuses on evaluating multiple metabolites in serum and urine from patients with ARDS. Information from this study may provide new biomarkers or targets for intervention, as the authors suggested that kynurenine is associated with mortality. The following is a synopsis of concerns raised with the data presented in the manuscript.
Major Concerns:
Point 1: It is unclear how the authors define “mortality.” Does the mortality have a time limit, such as 28-day or 90-day mortality?
Response 1: We thank the reviewer to point out this problem and we apologized for the lack of clarity to define mortality. We defined “mortality” as the hospital mortality. Our objective in this prospective study was to use metabolic profiling of amino acids and biogenic amines to examine the association between serial changes in metabolic profiles and “hospital mortality” among patients with ARDS.
We revised the title of the revised manuscript and added the word “hospital” mortality. We addressed the definition of hospital mortality in the second paragraph of Materials and Methods section in the revised manuscript as follows (marked with red text):
Hospital mortality refers to all-cause death during the hospital stay.
Point 2: What are the benefits of measuring kynurenine compared to existing models, such as the SOFA score?
Response 2: We appreciated the reviewer’s comment, and this is an excellent point of view. SOFA score is composed of six organ systems variables (respiration, coagulation, liver, cardiovascular, central nerve system, and renal) [1]; however, kynurenine is a single metabolite.
Predictive values of kynurenine and SOFA score at day 1 of ARDS onset on hospital mortality in our study were as follows: the area under the receiver operating characteristic (AUROC) of kynurenine at day 1 was 0.73 (p = 0.002), and AUROC of SOFA score at day 1 was 0.735 (p = 0.001). It seems that the predictive value of kynurenine at day 1 was not inferior to SOFA score at day 1 of ARDS onset.
The most common cause of death in ARDS is multiple organ failure [2], and SOFA score is a commonly used scoring system to reflect organ dysfunction. The kynurenine pathway of tryptophan metabolism plays a pivotal role in inflammatory responses and immune system activation (like during ARDS), and are associated with multiple organ failure.
Therefore, measuring kynurenine may be used as a valuable prognostic marker to predict disease progression/multiple organ failure or as a target for therapeutic interventions (like antioxidants) aimed at reducing morbidity and mortality in cases of ARDS in the future.
References:
- Singer, M.; Deutschman, C.S.; Seymour, C.W.; Shankar-Hari, M.; Annane, D.; Bauer, M.; Bellomo, R.; Bernard, G.R.; Chiche, J.D.; Coopersmith, C.M.; The Third International Consensus Definitions for Sepsis and Septic Shock (Sepsis-3). JAMA. 2016, 315, 801–
- Thompson, B.T.; Chambers, R.C.; Liu, K.D. Acute Respiratory Distress Syndrome. N. J. Med. 2017, 377, 562–572.
Point 3: Significant non-survivals have CKD and are immunocompromised hosts.
Response 3: We thank the reviewer’s comment to point out this problem. This prospective study was conducted in a single tertiary care referral center, and our ARDS patients may have more comorbidiites than other studies, which contributed to higher mortality (overall hospital mortality 52.2%). Hospital mortality in the LUNG SAFE study is 40.0% [1]. This no doubt limits the generalizability and reliability of our findings, and made our study external validation diffucult to perform.
We addressed this limitations in the eighth paragraph of Discussion section in the revised manuscript as follows (marked with red text):
This study was hindered by a number of limitations. First, all of the patients were from a single tertiary care referral center with small sample size, more comorbidities with higher hospital mortality, significant numbers of ARDS patients were on neuromuscular blockage, and therefore lacked external validation.
Reference:
- Bellani, G.; Laffey, J.G.; Pham, T.; Fan, E.; Brochard, L.; Esteban, A.; Gattinoni, L.; van, Haren. F; Larsson, A.; McAuley, D.F; et al. LUNG SAFE Investigators; ESICM Trials Group. Epidemiology, Patterns of Care, and Mortality for Patients With Acute Respiratory Distress Syndrome in Intensive Care Units in 50 Countries. JAMA. 2016, 315, 788–800.
Point 4: Kynurenine levels correlate well, mostly on day 1, but not in the later time points (3 and 7 days).
Response 4: We appreciated the reviewer’s comment. Predictive values of kynurenine on hospital mortality in this study were as follows: the area under the receiver operating characteristic (AUROC) of kynurenine at day 1, day 3 and day 7 were 0.73 (p = 0.002), 0.687 (p = 0.008), and 0.73 (p = 0.002), respectively. It revealed that the predictive value for mortality of kynurenine at day 1 was not inferior to later time potins (day 3, and day 7 of ARDS onset). Besides, early identification of the biological or clinical variables (like at day 1 of ARDS onset, not in the later time potins) to predict disease progression or mortality in ARDS is crucial. Therefore, we categorize patients according to plasma kynurenine levels at day 1.
Point 5: It is unclear whether kynurenine is truly an antioxidant, or whether this manuscript fits the overall theme of the journal. The article might be more suitable for a journal such as Critical Care Medicine, if the authors can justify the pitfalls of their study design.
Response 5: We thank the reviewer’s question and suggestion.
Patients with ARDS could experience severe oxidative stress from neutrophil recruitment and high inspired oxygen concentrations used. An increased generation of reactive oxygen species (ROS) during ARDS contributes to oxidative stress which overcome existing antioxidant defenses [1, 2], and plays an important role in the pathogenesis and progression in ARDS. The activation of kynurenine pathway was linked with an increased oxidative stress. Our study showed that kynurenine accumulation was found in ARDS patients, and nonsurvivors had significant higher values than survivors. It indicated that kynurenine accumulation in ARDS patients was associated with high oxidative stress and high mortality.
Antioxidants to overcome the excessive ROS generation could be a potential target for developing therapeutic strategies for ARDS. Although it is unclear whether kynurenine is truly an antioxidant, we hope that potentially therapeutic interventions like antioxidants targeting metabolic dysregulation of the kynurenine pathway could alleviate disease progression and reduce mortality in cases of ARDS in the future.
Therefore, we think that this manuscript fits the overall theme and is suitable for publication of “Antioxidants”
References:
- Quinlan, G.J.; Lamb, N.J.; Tilley, R.; Evans, T.W.; Gutteridge, J.M. Plasma hypoxanthine levels in ARDS: implications for oxidative stress, morbidity, and mortality. Am J Respir Crit Care Med. 1997, 155, 479–84.
- Kellner, M.; Noonepalle, S.; Lu, Q.; Srivastava, A.; Zemskov, E.; Black, S.M. ROS Signaling in the Pathogenesis of Acute Lung Injury (ALI) and Acute Respiratory Distress Syndrome (ARDS). Adv Exp Med Biol. 2017, 967, 105–137.
Point 6: Significant numbers of ARDS patients were on neuromuscular blockage, a practice less commonly used in the U.S. This raises the question if the data gathered in the study could be applied to a general environment.
Response 6: We thank reviewer to point out this problems. This prospective study was conducted in a single tertiary care referral center in Taiwan and significant numbers of the enrolled ARDS patients were on neuromuscular blockage probably due to higher disease severity, a practice less commonly used in the U.S. Only 21.7% of ARDS patients in the LUNG SAFE study were on neuromuscular blockage [1]. This issue limited the generalizability and reliability of our findings and made external validation diffucult to perform.
We addressed this limitations in the eighth paragraph of Discussion section in the revised manuscript as follows (marked with red text):
This study was hindered by a number of limitations. First, all of the patients were from a single tertiary care referral center with small sample size, more comorbidities with higher hospital mortality, significant numbers of ARDS patients were on neuromuscular blockage, and therefore lacked external validation.
Reference:
- Bellani, G.; Laffey, J.G.; Pham, T.; Fan, E.; Brochard, L.; Esteban, A.; Gattinoni, L.; van, Haren. F; Larsson, A.; McAuley, D.F; et al. LUNG SAFE Investigators; ESICM Trials Group. Epidemiology, Patterns of Care, and Mortality for Patients With Acute Respiratory Distress Syndrome in Intensive Care Units in 50 Countries. JAMA. 2016, 315, 788–
We thank the reviewer for valuable comments. Addressing them fully has significantly strengthened the manuscript.
Round 2
Reviewer 2 Report
The authors have addressed most of my remarks in the resubmission of their manuscript.
Yet, I am not convinced by the data that the non-survivors were indeed characterized by increased inflammation and oxidative stress and this was related to the alterations in the kynurenine pathway. The measured parameters that reflect inflammation and oxidative stress directly are not significantly different between the two groups, and do not relate to the kynurinine pathway in the regression models. So either the authors need to perform additional analysis on other markers of inflammation and more direct markers of oxidative stress, or rephrase their discussion section on this topic. The statement on the measurement of circulating cytokines is moreover still misleading in the discussion (to be interpreted as having been measured) and should be rephrased.
The addition of p-values in the supplementary table is helpful, but the comparison between healthy controls and ARDS patients is still lacking. The supplemental tables should be referred to earlier, together with the description of the volcano plots and heat maps.
The sentence on the categorization of patients with high and low kynurenine levels should be moved to section 3.3.
Ref 30 could be integrated better into the discussion - use these data and compare with own rather than just mentioning this study.
The explanation in the rebuttal on the use of urinary levels should be included in the discussion.
Author Response
The authors have addressed most of my remarks in the resubmission of their manuscript.
Point 1: Yet, I am not convinced by the data that the non-survivors were indeed characterized by increased inflammation and oxidative stress and this was related to the alterations in the kynurenine pathway. The measured parameters that reflect inflammation and oxidative stress directly are not significantly different between the two groups, and do not relate to the kynurinine pathway in the regression models. So either the authors need to perform additional analysis on other markers of inflammation and more direct markers of oxidative stress, or rephrase their discussion section on this topic. The statement on the measurement of circulating cytokines is moreover still misleading in the discussion (to be interpreted as having been measured) and should be rephrased.
Response 1: We thank the reviewer to point out this issue and and suggestions. We agree with the reviewer that nonsurvivors were not indeed characterized by increased inflammation and oxidative stress which was related to the kynurenine pathway directly by the results in our study. The measured parameters that reflect inflammation and oxidative stress like CRP was not significantly between two groups.
There are some markers for oxidative stress [1, 2]. However, circulating cytokines and markers of oxidative stress were not examined in this prospective study during 2017/02 to 2018/06.
We apologize again for the inappropriate statement on the measurement of circulating cytokines, and we did not perform cytokines analysis in this prospective study.
We add the description in the ninth paragraph of Discussion section in the revised manuscript as follows (marked with red text):
…Although cytokines analysis was not performed in this study to identify the potential metabolite-cytokine relationships….
We rephrase this topic in the tenth paragraph of Discussion section (the limitations) in the revised manuscript as follows (marked with red text) and added two references 41,42:
…Third, the measured parameters that reflected inflammation and oxidative stress were not significantly different between survivors and nonsurvivors, like CRP. The circulating cytokines (e.g., IL-6 or interferon-γ) were not measured in this study and further analysis is needed to examine the inflammatory status over the various timepoints. The markers of oxidative stress were also not examined [41, 42]. Therefore, we could not verify whether nonsurvivors were indeed characterized by increased inflammation and oxidative stress which was directly related to the alterations in the kynurenine pathway….
References:
- Marrocco, I.; Altieri, F.; Peluso, I. Measurement and Clinical Significance of Biomarkers of Oxidative Stress in Humans. Oxid Med Cell Longev. 2017, 2017, 6501046.
- Menzel, A.; Samouda, H.; Dohet, F.; Loap, S.; Ellulu, M.S.; Bohn, T. Common and Novel Markers for Measuring Inflammation and Oxidative Stress Ex Vivo in Research and Clinical Practice-Which to Use Regarding Disease Outcomes? Antioxidants (Basel). 2021, 10, 414.
Point 2: The addition of p-values in the supplementary table is helpful, but the comparison between healthy controls and ARDS patients is still lacking. The supplemental tables should be referred to earlier, together with the description of the volcano plots and heat maps.
Response 2: We thank the reviewer for the suggestion. We added two tables (Table S1 and Table S3) in the Supplementary Materials to compare the variables of plasma amino acids and biogenic amines between healthy controls and ARDS patients (at day 1, day 3, and day 7 after ARDS onset).
The supplemental tables were referred to the second paragraph of the Result section in the revised manuscript together with the description of the volcano plots and heat maps (marked with red text) (section 3.1).
Point 3: The sentence on the categorization of patients with high and low kynurenine levels should be moved to section 3.3.
Response 3: We thank the reviewer for the suggestion. The sentence on the categorization of patients with high and low kynurenine levels was moved to section 3.3.
Point 4: Ref 30 could be integrated better into the discussion - use these data and compare with own rather than just mentioning this study.
Response 4: We thank the reviewer for the suggestion to compare our results with the data of Reference 30.
We added the descriptions and comparisons of data between our study and Reference 30 in the third and fourth paragraphs of Discussion section in the revised manuscript as follows (marked with red text):
A recent study was conducted by Metwaly et al to identify metabolic fingerprints in ARDS and its subgroups (direct and indirect ARDS) and subphenotypes (hypoinflammatory and hyperinflammatory ARDS) [30]. It included ARDS patients (n = 108; median age 59 years; 28-day mortality 21.3%) and control group (ICU ventilated patients; n = 27; median age 57 years; 28-day mortality 12.5%). Most ARDS cases were secondary to pneumonia and sepsis, similar to our study. In longitudinal tracking of metabolites changes in the recovery group of ARDS (n = 43) at day 1, 7, 14 and 28 after ICU admission, it revealed that the levels of several metabolites appeared to move toward control levels within 7–14 days following ICU admission, which coincided with the time of clinical improvement. Kynurenine reached peak value at day 7 and slowly declined later in ARDS patients, but remained higher than control group at day 1 (fold change 9.9; p = 0.003), day 7, day 14, and day 28 after ICU admission. However, this above study did not evaluate the association between serial changes of metabolites and mortality.
In the current study, we enrolled healthy subjects as control group (n = 30; mean age 66.3 years) and ARDS patients (n = 69; mean age 65.2 years; hospital mortality 52.2%). We evaluated the association between serial changes in metabolic profiles and hospital mortality among patients with ARDS, and we didn’t compare the metabolic profiles of ARDS subgroups and subphenotypes as the previous study [30]. We also found that plasma kynurenine levels were significantly higher among ARDS patients on all sampling days (day 1, 3, and 7) than among control group (ARDS patients at day 1 vs healthy control: 25.1 ± 19.1 vs 3.7 ± 0.6 µM, p < 0.001). A stepwise increase in the mean concentration of plasma kynurenine was found among nonsurvivors from day 1 to day 7, and plasma kynurenine was the only metabolite that the mean value remained significantly higher among nonsurvivors than among survivors throughout the study period.…….
Point 5: The explanation in the rebuttal on the use of urinary levels should be included in the discussion.
Response 5: We thank the reviewer for the suggestion. We added the explanation in the rebuttal on the use of urine data to assess the association with hospital mortality and added one Reference 31.
We added the explanation in the fourth paragraph of Discussion section in the revised manuscript as follows (marked with red text):
However, the concentration of each urine metabolite was normalized to each urine sample’s corresponding creatinine level to compensate for urine volumes variations [31]. Twenty ARDS patients (29%) had chronic kidney disease and some ARDS patients experienced acute kidney injury with elevated creatinine, oliguria or even anuria in our study. This may make urine metabolites analysis less practical than serum metabolites and limit the numbers of patients available for statistical analysis. Therefore, the urine metabolites were not used to assess relations to hospital mortality in this study.
Reference:
- Bouatra, S.; Aziat, F.; Mandal, R.; Guo, A.C.; Wilson, M.R.; Knox, C.; Bjorndahl, T.C.; Krishnamurthy, R.; Saleem, F.; Liu, P.; et al. The human urine metabolome. PLoS One. 2013, 8, e73076.
We thank the reviewer for valuable comments. Addressing them fully has significantly strengthened the manuscript.